# Independent rediploidization masks shared whole genome duplication in the sturgeon-paddlefish ancestor

Anthony K. Redmond [●][1], Dearbhaile Casey[1], Manu Kumar Gundappa[2], Daniel J. Macqueen [●][2] & Aoife McLysaght [●][1] [✉]

Whole genome duplication (WGD) is a dramatic evolutionary event generating many new genes and which may play a role in survival through mass extinctions. Paddlefish and sturgeon are sister lineages that both show genomic evidence for ancient WGD. Until now this has been interpreted as two independent WGD events due to a preponderance of duplicate genes with independent histories. Here we show that although there is indeed a plurality of apparently independent gene duplications, these derive from a shared genome duplication event occurring well over 200 million years ago, likely close to the Permian-Triassic mass extinction period. This was followed by a prolonged process of reversion to stable diploid inheritance (rediploidization), that may have promoted survival during the Triassic-Jurassic mass extinction. We show that the sharing of this WGD is masked by the fact that paddlefish and sturgeon lineage divergence occurred before rediploidization had proceeded even half-way. Thus, for most genes the resolution to diploidy was lineage-specific. Because genes are only truly duplicated once diploid inheritance is established, the paddlefish and sturgeon genomes are thus a mosaic of shared and non-shared gene duplications resulting from a shared genome duplication event.

Ancient WGD events have occurred across the tree of life and are especially well studied in plants[1–4], yeast[5,6], and vertebrates[7–18]. These events are often hypothesised to have facilitated evolutionary success through provision of the raw genetic materials for phenotypic innovation and species diversification[1,19–23]. A key evolutionary process after WGD is rediploidization—the transition of a polyploid, usually tetraploid, genome to a more stable diploid state[1,7,9,14,22–25]. Importantly in this context, WGD events are derived from either hybridisation of two different parent species (allopolyploidisation) or from doubling of the same genome at the intra-species/individual level (autopolyploidisation), each with distinct cytogenetic outcomes. Classically, the non-homologous chromosomes of new allopolyploids preferentially pair bivalently during meiosis, whereas the four homologous chromosomes of new autopolyploids take a multivalent formation[1,9,14,22,24]. This

results in ongoing homologous recombination, and hence gene conversion and homogenisation, across the four allelic copies at each locus. Suppression of recombination, probably achieved through chromosomal rearrangements and other mutations, is a necessary step to rediploidize these genes into two distinct (bivalent) ohnolog loci (WGD-derived duplicate genes) from tetraploid alleles. It is only then that uninterrupted sequence and functional divergence can occur between ohnolog pairs[24]. The rediploidization process thus uncouples the genome duplication process from the gene duplication process in autopolyploids, as it is only once rediploidization has occurred that the locus can be considered duplicated.

Substantial evidence arising from studies of the ancestral salmonid WGD[18,24,26], and ancestral teleost WGD[13,27,28], indicates that autopolyploid rediploidization can be temporally protracted, occurring

[1]Smurfit Institute of Genetics, Trinity College Dublin, Dublin, Ireland. [2]The Roslin Institute and Royal (Dick) School of Veterinary Studies, University of Edinburgh, Edinburgh, UK. [✉]e-mail: aoife.mclysaght@tcd.ie

asynchronously across the genome over tens of millions of years[13,18,24,26–28]. Major implications arise from the accompanying delay in any evolutionary processes that depend on ohnolog genetic divergence[24]. These include well-established models of functional evolution after gene duplication, e.g. sub-/neo-functionalization[29,30], and models of reproductive isolation involving reciprocal loss of ohnologs in sister lineages[31]. Furthermore, if speciation occurs before rediploidization has completed in descendent lineages of the same WGD, rediploidization may occur independently in these daughter lineages (lineage-specific rediploidization) in some genomic regions[24]. This in turn allows for ohnolog pairs to independently evolve divergent regulatory and functional trajectories in each lineage, potentially in response to lineage-specific selective pressures[24]. Ohnologs with this history have been described as following the LORe (Lineage-specific Ohnolog Resolution) model[24].

Although noted as a potentially important evolutionary process after several WGDs[4,8,12,24,27,32–34], it remains unclear whether asynchronous and lineage-specific rediploidization is a general feature after autopolyploid WGD outside the teleost clade. The past few years have seen the generation of high-quality reference genomes from multiple non-teleost ray-finned fish lineages[7,8,35,36], which share the two ancient rounds of WGD common to all jawed vertebrates[9,11,37], but lack the teleost-specific WGD event[7,8,35,36,38,39]. These species typically have slowly evolving genomes, enabling more reliable inference of the ancestral state of bony vertebrates than teleosts, while providing outgroups to understand the impact of the teleost-specific WGD event[7,8,35,36,39]. Among these newly available genomes are chromosome-scale assemblies for the sterlet sturgeon (*Acipenser ruthenus*)[7] and American paddlefish (*Polyodon spathula*)[8], which each have experienced WGD in their evolutionary histories.

Multiple WGD events have occurred during sturgeon evolution, but only one, thought to be shared by all sturgeons, is present in the sterlet sturgeon's history[7,40–42]. On the other hand, American paddlefish is the only extant paddlefish[43], and is suggested to have undergone a single WGD event[8,35,44–46]. Despite being sister lineages, together representing extant Acipenseriformes, previous analyses have consistently rejected a shared ancestral WGD in favour of independent WGD events (Fig. 1)[8,35,38,44,45]. Efforts to date these WGDs have produced incongruent results, with estimates ranging from 21.3 Ma[38], 51 Ma[35], and 180 Ma[7] for the sturgeon WGD, and 41.7 Ma[44], ~50 Ma[8], and 121 Ma[35] for the paddlefish WGD. Although some authors have suggested that asynchronous rediploidization may contribute to the observed incongruence across studies[8,44], this process has been ignored when dating these WGDs. Furthermore, the potential for genome-wide lineage-specific rediploidization (e.g.[13,24,26]) to mask a shared WGD event has never been formally proposed or tested in any lineage.

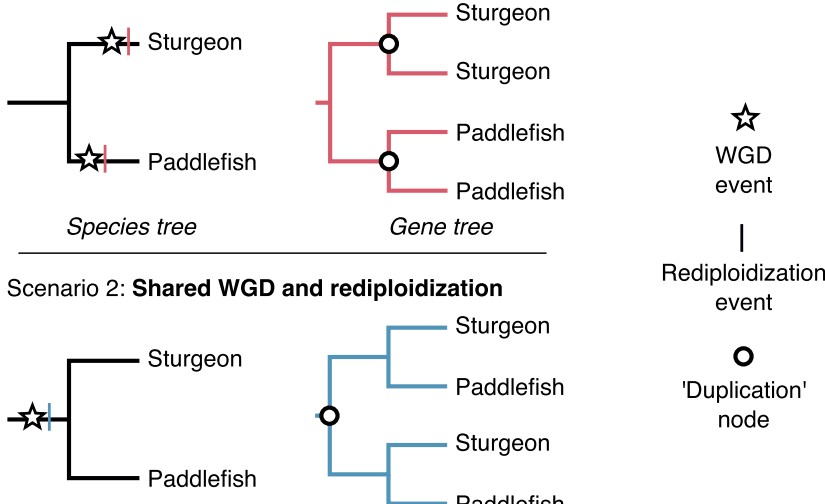

**Fig. 1 | Scenarios of WGD and rediploidization timing relative to the Sturgeon-Paddlefish divergence and their expected topologies for ohnolog-pair gene trees.** Scenario 1 is the widely accepted hypothesis of independent WGD events in the sturgeon and paddlefish lineages. Scenario 2 is a shared ancestral WGD with complete rediploidization prior to lineage divergence. Scenario 3 extends Scenario 2 by considering the possibility of speciation happening during a prolonged, asynchronous rediploidization process following a shared WGD event. In this case genes rediploidizing prior to speciation will follow the gene tree expected under scenario 2 while those rediploidizing after speciation (i.e. lineage-specific rediploidization) will follow the gene tree expected under Scenario 1. This is distinguishable from independent small-scale duplication using the expectation that ohnolog pairs largely retain ancestral collinearity between non-overlapping duplicate chromosomal regions. Rediploidization events and associated gene trees after the sturgeon-paddlefish speciation are shown in red, those preceding speciation are shown in blue.

Here, adopting a phylogenomic approach, we reconsider the timing of WGD(s) relative to the sturgeon-paddlefish divergence, accounting for the possibility of lineage-specific rediploidization after a shared WGD. Taking care to distinguish our results from phylogenetic error, and drawing on conserved synteny, we provide strong evidence for a single ancestral autopolyploidy occurring close to the Permian-Triassic extinction event. This was followed by extensive lineage-specific rediploidization, resulting from the ancestral acipenseriform genome remaining predominantly tetraploid at the time of sturgeon-paddlefish divergence, a genomic condition that may have promoted survival of the lineage through the Triassic-Jurassic mass extinction.

## Results

### Recovery of ohnolog pair subsets diverging before and after speciation

Past studies assessing the timing of WGD(s) in acipenseriform history have sought a single consensus ohnolog divergence time relative to speciation[8,35,44]. This approach considers two scenarios as plausible: (1) if a plurality of ohnolog gene trees recover independent duplication nodes after the sturgeon-paddlefish divergence then sturgeons and paddlefish are assumed to have undergone independent WGDs (currently accepted hypothesis) (Fig. 1, Scenario 1); and (2) if a plurality of ohnolog gene trees show duplication nodes predating the sturgeon-paddlefish divergence then a single ancestral WGD event can be assumed (typically rejected hypothesis) (Fig. 1, Scenario 2). These interpretations implicitly assume all ohnologs share the same rediploidization timing. Here, we consider a third plausible scenario, as previously observed to have occurred after the ancestral salmonid and teleost WGD events[13,18,24,26]: (3) a shared WGD followed by a prolonged rediploidization process that starts before but continues after speciation. This predicts the presence of two distinct subsets of ohnolog gene trees, one with duplication nodes prior to the sturgeon-paddlefish divergence and the other with independent duplication nodes after speciation (Fig. 1, Scenario 3).

To distinguish between these scenarios, we built on a set of high confidence ohnologs previously identified in the sturgeon genome[7], integrating extensive additional phylogenetic and syntenic evidence. Specifically, we incorporated a broad sampling of predicted proteomes from jawed vertebrate genomes, including from newly available non-teleost ray-finned fish. This allowed us to define 5,439 protein-coding gene families containing high confidence ohnolog pairs in both sturgeon and paddlefish. Analysing maximum likelihood gene trees for each family we found that the gene tree harbouring independent duplication nodes was the most common topology (hereafter: 'PostSpec', for Post-Speciation duplication node, as in Scenario 1 and Scenario 3-right), being recovered 2074 times (38.13% of all trees; Fig. 2A). The alternative ohnolog pair topology with a shared duplication node ('PreSpec' for Pre-Speciation, as in Scenario 2 and Scenario 3-middle) was recovered 1448 times (26.62% of all trees; Fig. 2A). The remaining gene trees (1917, 35.25%; 'Other' for topologies other than PostSpec or PreSpec) failed to recover either of these topologies.

The frequent recovery of the PostSpec topology likely explains why previous studies inferred that sturgeons and paddlefish underwent independent WGDs[8,35,38,44,45] (Scenario 1; Fig. 1), however the high prevalence of the PreSpec topology and 'Other' topologies in our analyses requires explanation. Firstly, the frequent recovery of both PreSpec and PostSpec topologies is consistent with a shared WGD followed by prolonged rediploidization extending past the sturgeon-paddlefish speciation (Scenario 3; Fig. 1). In this case, the PreSpec and PostSpec topologies map directly to the ancestral and lineage-specific ohnolog resolution models (dubbed 'AORe' and 'LORe') previously described in salmonids[24]. However, given the high proportion of inferred 'Other' topologies it is important to consider whether the

variability in ohnolog divergence time estimates is impacted by phylogenetic error. Similarly, notwithstanding our efforts to define a set of high confidence ohnologs, it is also important to ensure that small-scale duplication events do not drive recovery of either the PreSpec or PostSpec topologies.

### Variation in ohnolog divergence time is not a product of phylogenetic error

To determine the possible impact of phylogenetic error on our findings, we considered factors that may have influenced the initial tree topologies recovered. The critical branching pattern informing our competing hypotheses is the clade in each rooted gene family tree comprising the paddlefish and sturgeon ohnolog pairs (i.e. a four-gene subtree). First, we considered the recovery of three broad topology categories (PostSpec, PreSpec, 'Other'; Fig. 2A) in light of the 15 possible rooted topologies that a four-taxon tree can take. One of these 15 topologies maps to PostSpec, two to PreSpec, and the remaining 12 to 'Other' topologies (Fig. 2A). However, these 'Other' topologies naturally fit into two categories; 'PostSpec-like', and 'PreSpec-like', each of which were recovered at a frequency in line with their closest main topology (i.e. PostSpec/PreSpec), and require only a single branch change (which could be explained by a minor inference error) to be recovered as PostSpec or PreSpec, respectively (Fig. 2A). Further, such minor topology differences would be indistinguishable from their closest main topology (PostSpec/PreSpec) if the trees were unrooted (Fig. 2A, B). The PostSpec and PreSpec topologies, but not 'Other' topologies are recovered more frequently than would be expected by random chance (i.e. assuming a 1 in 15 chance for any given topology; Fig. 2A). This indicates a strong signal for PreSpec and/or PostSpec but not for 'Other' topologies.

To confirm the strength of signal supporting each topology we performed unrooted Approximately Unbiased (AU)-tests[47] on the sturgeon-paddlefish ohnolog pair subtrees, considering the three possible unrooted topologies of a four-taxon tree; one PostSpec-type (the unrooted equivalent of both PostSpec and PostSpec-like), and two PreSpec-type (the unrooted equivalent of both PreSpec and PreSpec-like) (Fig. 2B). The results indicate that ohnolog pairs recovering the rooted PostSpec and PreSpec topologies are more robust. Specifically, they frequently reject the unrooted alternative topology type; whereas those that recovered the rooted PostSpec-like and PreSpec-like 'Other' topologies reject the unrooted alternative topology type less frequently (Fig. 2B). Although the PreSpec/PostSpec-like 'Other' datasets are more indecisive, the matching unrooted topology type is almost never rejected in favour of the unrooted alternative topology across any of the four rooted topology sets (Fig. 2B). Rather than any single topology providing a consensus, these results are consistent with significant support for both the PostSpec and PreSpec topologies within our wider ohnolog pair dataset, and with the 'Other' topology, being derived from less informative gene family alignments.

As an additional test of tree robustness we assessed the impact of filtering trees based on increasingly stringent branch support cut-offs (i.e. Ultrafast Bootstrap [UFBoot][48]) within the sturgeon-paddlefish ohnolog pair subtree. As stringency increases, we observe a drop out of all tree topologies (Fig. 2C). However, this is most severe for 'Other' topologies, which are rarely recovered at high stringency, being recovered over 40 times less often than random at the strictest cut-off (UFBoot = 100%) (Fig. 2C). On the other hand, PostSpec and PreSpec topologies were recovered much more often than expected at random, regardless of UFBoot cut-off, with the PreSpec topology overtaking PostSpec as the most frequently recovered topology at UFBoot ≥ 95% (Fig. 2C). This further confirms a strong, non-random signal for both the PostSpec and PreSpec topologies in our full ohnolog pair gene family dataset.

Next, as a proxy for whether we can expect the sturgeon-paddlefish subclade to be recovered accurately, we compared the

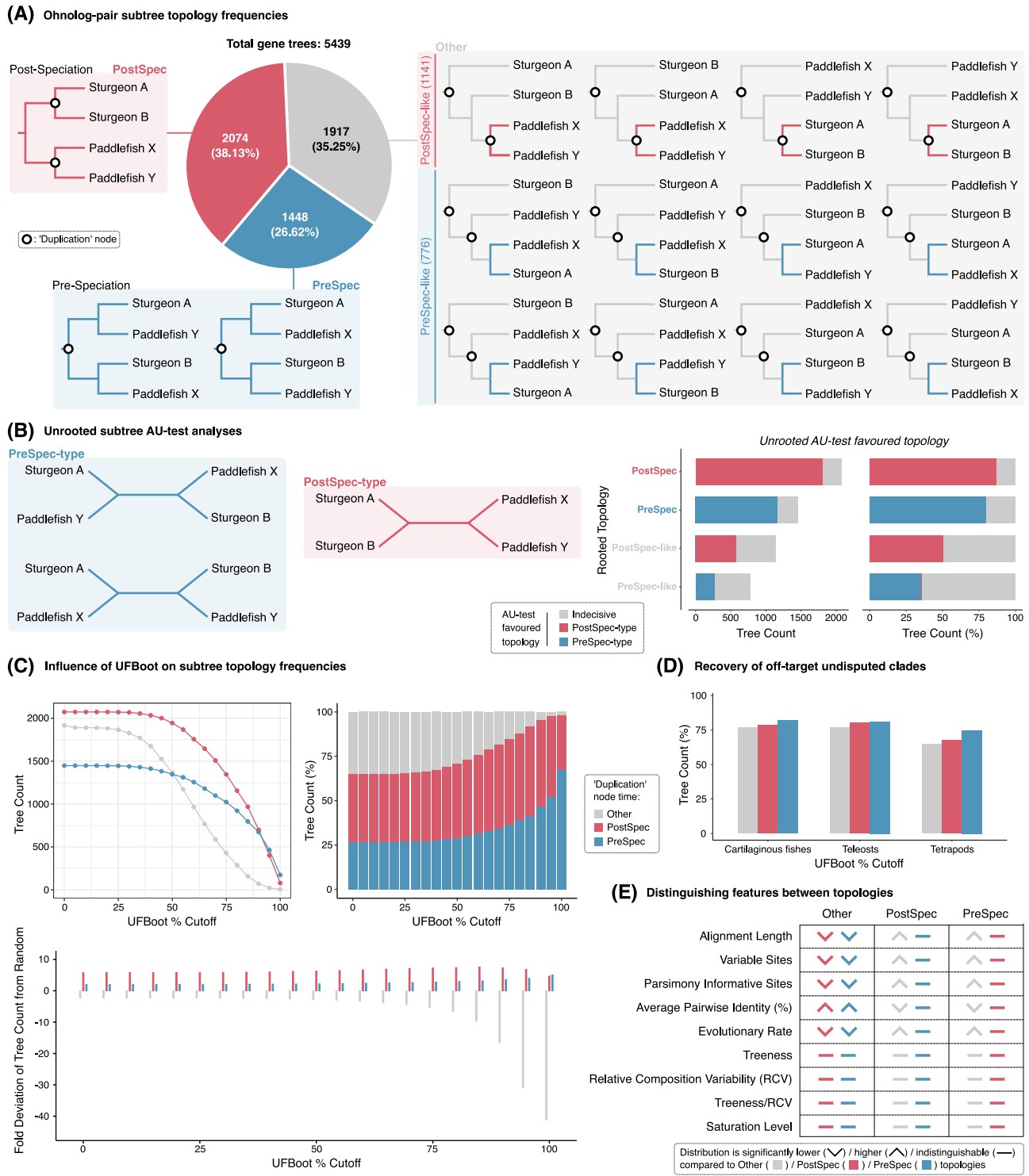

**Fig. 2 | Ohnolog pair gene tree topologies and investigation of possible sources of phylogenetic error. A** Categorisation of the 15 possible rooted sturgeon-paddlefish subtrees with duplication nodes coming before ('PreSpec') or after ('PostSpec') these species diverged, and 'Other' trees that only partially match one of these scenarios (either, 'PreSpec-like', or 'PostSpec-like'). The pie chart quantifies the relative frequency at which each topology was recovered. **B** Three possible unrooted sturgeon-paddlefish subtrees (two 'PreSpec-type', left; and one 'Post-Spec-type', centre), and Approximately Unbiased (AU)-test (right) of tree reliability to determine how frequently datasets from each category of rooted subtree described in part (**A**) can decisively reject a given unrooted topology category type and thereby favour the other. **C** Rooted subtree topology category (broken down at

the 'Other', 'PostSpec', 'PreSpec' level) count (top left), percentage (top right), and fold deviation of the tree count per category from random expectations (i.e. as estimated if each of the 15 rooted trees were recovered equally frequently; bottom) under increasingly strict UFBoot percentage cut-offs, such that both UFBoot percentages in a given subtree must be greater than or equal to the cut-off for that tree to be retained. **D** Percentage of trees fitting each sturgeon-paddlefish subtree category that recover other key undisputed clades. **E** Summary of significant differences across sequence alignment, modelling, and tree-based statistics for each subtree category (Supplementary Fig. 1 provides violin/box plots with *p* values). Source data are provided as a Source Data file. Raw alignments, gene trees, and gene tree parsing code are provided on figshare[106].

ability of gene trees supporting each sturgeon-paddlefish ohnolog pair topology to recover other well-accepted clades[49,50], i.e. cartilaginous fishes, tetrapods, teleosts (Fig. 2D). If a gene tree fails to recover known, well-supported clades, it may be indicative of generally low phylogenetic signal. Although no major differences were observed between the three topology categories, PreSpec trees consistently performed best, and 'Other' the worst (Fig. 2D).

Having confirmed the robustness of the phylogenetic signal, we sought to test whether systematic errors might drive a consistent and misleadingly strong signal for either of PostSpec and PreSpec topologies. We analysed a variety of statistics[51,52] at sequence alignment, modelling and tree topology levels (Fig. 2E, Supplementary Fig. 1). 'Other' tree topologies derive from shorter multiple sequence alignments with fewer substitutions per site (i.e. higher average pairwise identity, slower evolutionary rate[53], and fewer variable and parsimony informative sites) than PostSpec or PreSpec topologies (Fig. 2E, Supplementary Fig. 1). The combination of these factors presumably limits phylogenetic signal, consistent with the idea that 'Other' topologies result from weakly-supported, minor phylogenetic errors. We observed no significant differences across any statistics considered between the PreSpec and PostSpec topologies (Fig. 2E, Supplementary Fig. 1). Importantly, although they have more substitutions per site than 'Other' trees, PostSpec and PreSpec datasets do not show signs of being more susceptible to systematic errors, having a comparable balance of substitutions on internal and external tree branches (treeness), and similar compositional variability and substitutional saturation levels to 'Other' datasets (Fig. 2E, Supplementary Fig. 1)[52,54,55]. Lastly, site-heterogeneous models can help to alleviate systematic error-induced branching artefacts in phylogenetic analysis[56]. Testing their use on all gene families that had maximal support values (UFBoot = 100%) within the sturgeon-paddlefish ohnolog pair subclade never resulted in a topology change, while support values never dropped below UFBoot = 97% (Supplementary Fig. 2).

Together these analyses indicate that neither the PostSpec or PreSpec topologies derive from error, indicating strong and reliable support for ohnologs diverging both before and after the sturgeon-paddlefish divergence, while suggesting that 'Other' tree topologies are often a product of comparatively limited phylogenetic signal.

## Some paddlefish ohnologous regions form a single assembly sequence

Thousands of additional genes are annotated in the sturgeon genome compared to paddlefish[7,8,35]. To assess the impact of this on our findings we analysed the set of sturgeon ohnolog pairs for which a single ortholog is annotated in paddlefish. In these gene trees the sturgeon-paddlefish ohnolog subtree consists of only three genes. As such, there are only three possible rooted topologies: two being similar to PreSpec and one to PostSpec (Supplementary Fig. 3). We label these 'PreSpec-type' and 'PostSpec-type' as it is not possible to rule out these trees instead forming 'Other' topologies were a second paddlefish sequence to be introduced.

Among these apparently single-copy paddlefish genes, there is a greater proportion of PostSpec-type trees (1836 trees, ~82%) to PreSpec-type (397 trees, ~18%) than recovered for the main data above (Supplementary Fig. 3). As our earlier analyses appear to rule out a phylogenetic bias towards any given topology, two plausible explanations arise. Either late-rediploidising ohnologs have a greater tendency to revert to singleton status or some highly similar duplicate genomic regions are collapsed into a single assembly sequence. The challenges in distinguishing such regions of high sequence similarity has been noted in analysis of the sturgeon genome[7], and the potential for both collapse and maintenance of tetraploidy has been noted in salmonids, particularly for the European grayling (*Thymallus thymallus*)[26,57].

If a small proportion of duplicated paddlefish loci are still undergoing tetrasomic inheritance or are artefactually collapsed into a single locus, then this should be apparent as regions with twice the sequencing depth compared to the rest of the genome. This is because reads derived from two distinct genomic locations should map to collapsed regions[26,57]. To test this, we examined the distribution of genome sequencing read depth for each paddlefish gene within a retained ohnolog pair, as well as separately examining read depth for single-copy paddlefish genes with either PostSpec-type or PreSpec-type topologies (i.e. single-copy paddlefish loci where sturgeon retains both ohnologs).

The main density peak when plotting per-gene sequencing read depth for the single-copy gene sets is comparable to each of the two-copy paddlefish ohnolog pair genes. This implies that gene loss in paddlefish, rather than maintenance of tetraploidy or assembly collapse, explains most of these cases (Supplementary Fig. 3). However, we find a smaller, but clear, double coverage peak for single-copy paddlefish genes. This accounts for a substantially greater proportion of PostSpec-type than PreSpec-type single-copy genes (Supplementary Fig. 3). This is consistent with the expectation that more recently rediploidized ohnolog pairs (PostSpec and PostSpec-type) will have the highest sequence similarity and be more prone to assembly difficulty and artefacts.

## Conserved synteny supports a shared WGD followed by prolonged and asynchronous rediploidization

The autopolyploid rediploidization process is thought to involve numerous physically independent genomic rearrangement events across the genome[24]. Assuming that these rearrangements are not restricted to single genes[24,26], and that subsequent rearrangements are not extensive, large blocks of neighbouring genes sharing common rediploidization histories should be visible as largely non-overlapping syntenic blocks on different chromosomes, and present in both lineages. This is not unlike the history of suppression of recombination during the evolution of mammalian sex chromosomes, where genome rearrangements are associated with the onset of locus divergence on the X and Y and resulted in contiguous strata of genes that share an X-Y divergence time[14,58]. If our phylogenetic results arise from a shared WGD followed by a prolonged rediploidization spanning both the shared and lineage-specific branches, such divergence-time-stratified synteny blocks should be highly evident, especially considering that acipenseriform genomes evolve slowly and show limited reorganisation after WGD[7,8,35].

Plotting ohnolog pairs within and across the sturgeon and paddlefish genomes revealed that ohnologs from both the PreSpec and PostSpec categories (Fig. 2) are not randomly distributed along the genome. Instead ohnolog pairs with shared divergence dates relative to speciation (PreSpec or PostSpec) are found in syntenic blocks along large uninterrupted sections of chromosomes (and possibly even entire small chromosomes) (Fig. 3). For example, long PreSpec synteny blocks are conserved across both genomes on the six largest chromosomes (which form three WGD-derived pairs across both species[7,8]; Fig. 3C), making small-scale segmental duplication prior to lineage-specific WGD an implausible explanation for these topologies, and adding further support to the hypothesis that they reflect true evolutionary signal stemming from WGD. In all, these observations are parsimoniously explained by a single ancestral WGD followed by extensive ancestral and lineage-specific rediploidization in sturgeon and paddlefish evolution.

This result stands even when examining only the macrochromosomes (>40 Mb) (Supplementary Fig. 4) and for ohnolog trees with progressively stricter statistical support (i.e. UFBoot cut-off scores of ≥50%, ≥75%, and 100%) (Supplementary Fig. 5). Meanwhile, ohnolog trees from the 'Other' category, and from the sturgeon ohnolog pairs with only one paddlefish sequence, tend to occupy genomic regions

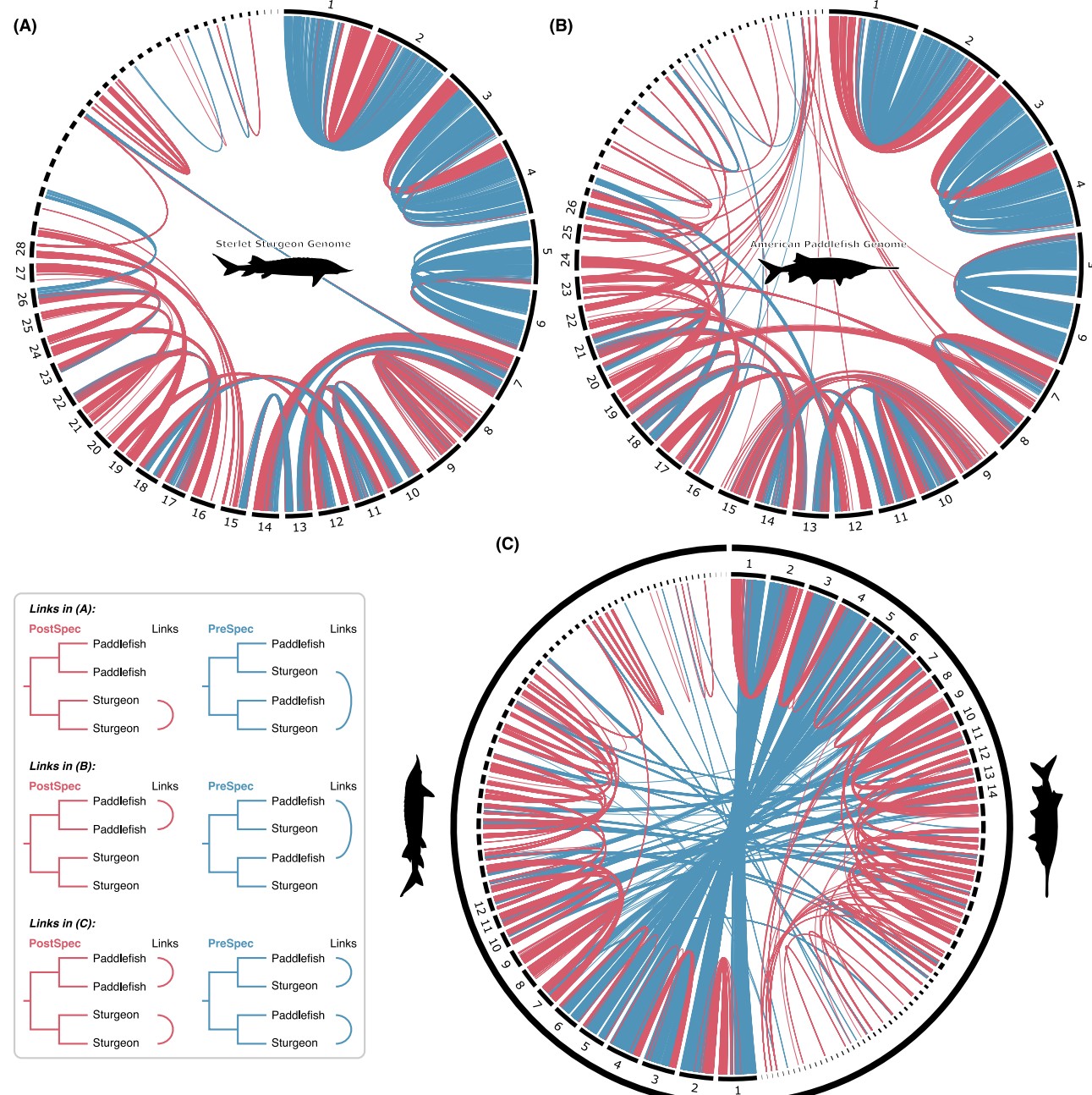

**Fig. 3 | Synteny patterns of 'PreSpec' and 'PostSpec' ohnolog pairs in the paddlefish and sturgeon genomes.** Circos plots of the sterlet sturgeon genome (**A**) and the American paddlefish genome (**B**) showing the chromosomal locations of ohnolog pairs, with links coloured according to the PreSpec (blue) or PostSpec (red) tree topology. Microchromosomes <20 Mb are not labelled. **C** Circos plot of ohnolog-pairs in both the sturgeon and paddlefish genomes, with intra-specific PostSpec links (red) and inter-specific PreSpec links (blue). Only macrochromosomes >40 Mb from each species are labelled. Source data are provided as a Source Data file.

harbouring genes with the most similar of the two main topologies (i.e. PreSpec-like and PreSpec-type alongside PreSpec, and PostSpec-like and PostSpec-type alongside PostSpec; Supplementary Figs. 3 and 6) as expected if these topologies primarily arise from minor errors.

### Early and late rediploidizing ohnologs fit distinct $K_s$ distributions

The distribution of synonymous substitutions per site ($K_s$) between paralogous genes within a genome is often applied as the basis for detecting WGD events and estimating their timing relative to speciation[59–61]. This is typically achieved through statistical modelling and/or simple visual identification of distribution peaks in $K_s$ plots.

Where asynchronous (including lineage-specific) rediploidization has occurred it naturally follows that a strong, single $K_s$ peak cannot be expected—rather the signal of ohnolog $K_s$ values, in so far as it can be taken as a proxy for time, will be more diffuse resulting in a broader, flatter peak or series of low peaks spanning the $K_s$ values across the rediploidization period[26].

With this in mind, and to further explore the differences between our PreSpec, PostSpec, and 'Other' topology datasets, we calculated pairwise $K_s$ values for species-specific ohnolog pairs from the PreSpec, PostSpec, and PreSpec- and PostSpec-like datasets, as well as all these combined (rows labelled 'All' in Fig. 4). We also calculated pairwise $K_s$ values for two sturgeon-paddlefish ortholog pair datasets for

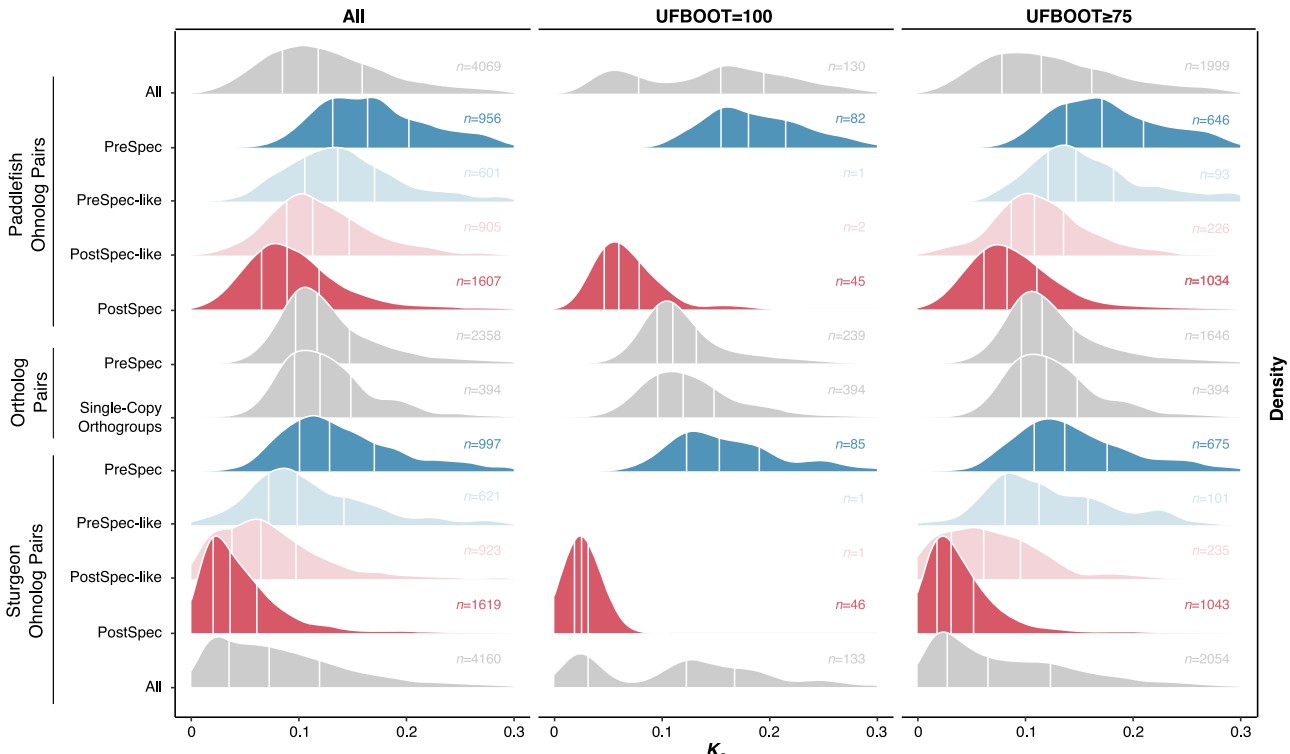

**Fig. 4 | $K_s$ value ridgeline density plots for distinct ohnolog and ortholog pair datasets at varying UFBoot cut-off percentages.** Species-specific ohnolog pair $K_s$ value densities are plotted for each of the four major topology categories (i.e. PostSpec, PreSpec, PostSpec-like, and PreSpec-like) for intra-species ohnolog pair data. For comparison we also plotted two sets of paddlefish-sturgeon ortholog pairs: (i) Single-Copy Orthogroups−sturgeon and paddlefish sequences present in the single copy genes identified in all species by OrthoFinder, and (ii) PreSpec Ortholog pairs−these derive from each ohnolog in the PreSpec topology such that a single PreSpec topology contributes two ortholog pairs whose divergence matches the sturgeon-paddlefish speciation. White vertical lines split each distribution into four quantiles, and the number of ohnolog/ortholog pair $K_s$ values ($n$) underlying each distribution is also displayed per dataset. $K_s$ values ≥ 0.3 were excluded, as well as pairs where a coding sequence was flagged as potentially problematic (e.g. early stop codon) by the wgd software tool used to calculate $K_s$ values. Source data are provided as a Source Data file.

comparison−single-copy orthogroups and PreSpec orthologs. We recover a clear distinction between $K_s$ distributions, with the PostSpec ohnolog pair data having the lowest $K_s$ peak, the PreSpec ohnolog pairs having the highest $K_s$ peak, and the ortholog $K_s$ peak falling intermediate to these (Fig. 4). This higher $K_s$ of early rediploidizing ohnologs (i.e. PreSpec), and lower $K_s$ of late rediploidizing ohnologs (i.e. PostSpec), compared to ortholog $K_s$ values is consistent with our predictions, and lends further support to the scenario of a shared WGD followed by an asynchronous rediploidization process.

Interestingly, PostSpec-like and PreSpec-like pair $K_s$ peaks also fall intermediate to the PostSpec and PreSpec pair $K_s$ peaks, and on a gradient with PostSpec-like always lying closer to PostSpec and PreSpec-like always closer to PreSpec (Fig. 4). We earlier considered whether these 'Other' topologies might primarily derive from branching errors due to limited phylogenetic signal. If, as suggested by the $K_s$ analysis, these gene pairs diverged close in time to the speciation events, then we would anticipate limited phylogenetic signal to distinguish speciation and duplication nodes within ohnolog pair subtrees, as indeed we observe for 'Other' topologies which are very rarely strongly supported (Fig. 2C).

By comparison to the $K_s$ densities separated by topology category we found wider, flatter $K_s$ density peaks when analysing all ohnolog pairs from each species, especially when not filtering based on UFBoot support (Fig. 4). When filtering for 100% UFBoot support a clear bimodal distribution can be observed (Fig. 4), likely representing strongly supported PreSpec and PostSpec rediploidization outcomes. These findings follow our proposed expectations for $K_s$ analyses under asynchronous rediploidization but are at odds with the traditional

predictions for detection and timing of WGD events with $K_s$. In this context our results are restricted to ohnolog sets rather than all duplicates, as is usually the case when such analyses are performed, meaning the signal would likely be even more diffuse in an ab initio WGD detection scenario.

Ohnolog pair $K_s$ peaks at higher values (i.e. greater divergence) are consistently recovered for paddlefish than for sturgeon (Fig. 4). This may be indicative of a faster rate of sequence evolution in paddlefish. To explore this, we tested whether paddlefish sequences had longer branch lengths on average than their sturgeon orthologs in PreSpec and Single-Copy Orthogroup gene trees and found there to be a significant difference for PreSpec ortholog pairs only (Supplementary Fig. 7). This suggests that the rate of sequence evolution in paddlefish may be faster than that in sturgeon. Alongside the apparent collapse of some highly similar ohnologous regions in the paddlefish genome (which would artefactually reduce the number of paddlefish ohnolog pairs with low $K_s$ values), this likely explains the trend towards higher $K_s$ values in paddlefish.

In all, these results are clearly consistent with asynchronous rediploidization masking a shared ancestral WGD in the sturgeon-paddlefish ancestor. Moreover, they indicate that approaches reliant on identifying and dating of WGD events by detecting a single $K_s$ peak may be compromised by protracted rediploidization periods.

## A Permian-Triassic lower bound for the sturgeon-paddlefish WGD

Asynchronous rediploidization temporally separates ohnolog divergence from WGD, obscuring the dating of autopolyploidy events[24,26,62].

Our gene tree and $K_s$ analyses suggest this is likely a major factor influencing the disparate array of dates previously proposed for WGD in sturgeon and paddlefish. Although imperfect, the most reliable lower bound estimate for the ancestral sturgeon-paddlefish WGD event can be estimated from ohnolog pairs that rediploidized prior to the sturgeon-paddlefish divergence, as these will have diverged closer in time to the WGD event[62]. With this in mind, to estimate a lower bound timing for the WGD we took a Bayesian phylogenomic approach[63] using concatenated ohnolog pairs[32,62] based on the set of 81 gene trees that maximally supported shared WGD and did not include duplicates in other species. We analysed five distinct datasets, always including all 81 gene families but randomly shuffling ohnologs from a pair for arbitrary assignment as the A or B copy for concatenation, to avoid bias and assess robustness of results to alternative concatenations[62]. We analysed these datasets with an autocorrelated relaxed molecular clock[64], and used the site-heterogeneous CAT-GTR substitution model[65], and considered two fossil calibration strategies (Supplementary Data 1). The first incorporating fossil evidence to apply upper and lower bound calibrations to key divergences between the major ray-finned fish lineages, and the second leaving most of these divergences uncalibrated given the difficulty in phylogenetically placing important Paleozoic ray-finned fish fossils (Supplementary Data 1, Fig. 5)[66].

In the first case (i.e. including all calibrations) we infer a divergence time of ~171.6 Ma (average mean of all five random concatenations [AVG-5RC]; 95% credibility interval range [CIR]: ~124.1–203.3 Ma) for the split of sturgeons and paddlefish (i.e. crown Acipenseriformes; considering both ohnolog pairs) (Supplementary Data 1, Fig. 5A, Supplementary Fig. 8). Following the split of Chondrostei (of which sturgeons and paddlefish are the only living representatives and form crown Acipenseriformes) and Neopterygii (i.e. crown Actinopteri) - 367.8 Ma (AVG-5RC; 95% CIR: ~360.6–374.8 Ma), we estimate a lower bound for the shared sturgeon-paddlefish WGD at ~254.7 Ma (AVG-5RC; 95% CIR: ~207.1–289 Ma) (Supplementary Data 1, Fig. 5A, Supplementary Fig. 8). Interestingly, this mean Bayesian estimate for the timing of the ancestral sturgeon-paddlefish WGD lower bound sits close to the Permian-Triassic (P-Tr) boundary mass extinction event ~251.9 Ma.

However, our second set of analyses, employing fewer ray-finned fish calibrations, places the WGD lower bound at ~241.8 Ma (AVG-5RC; 95% CIR: ~202.9–273.4 Ma) (Supplementary Data 1, Fig. 5B, Supplementary Fig. 9). In line with this, other uncalibrated divergences are also similarly shifted towards the present, including crown Neopterygii (Teleostei and Holostei; from 294.5 Ma [AVG-5RC; 95% CIR: ~270.8–319.9 Ma] to 278.5 Ma [AVG-5RC; 95% CIR: ~256.9–304.6 Ma]) and crown Holostei (i.e. gars and bowfin; from 276.9 Ma [AVG-5RC; 95% CIR: ~251.4–306.4 Ma] to 263.4 Ma [AVG-5RC; 95% CIR: ~240.6–291.6 Ma]), or are more dramatically shifted towards the present, such as crown Actinopterygii (all extant ray-finned fishes; from 378 Ma [AVG-5RC; 95% CIR: ~372.1–384 Ma] to 349.5 Ma [AVG-5RC; 95% CIR: ~329.4–370.9 Ma]), and crown Actinopteri (from 367.8 Ma [AVG-5RC; 95% CIR: ~360.6–374.8 Ma] to 340.2 Ma [AVG-5RC; 95% CIR: ~320.3–361.7 Ma]) (Supplementary Data 1, Fig. 5, Supplementary Figs. 8 and 9). The sturgeon-paddlefish (crown Acipenseriformes) divergence is also very slightly more recent in this analysis at ~167.5 Ma (AVG-5RC; 95% CIR: ~123.4–202.1 Ma) (Supplementary Data 1, Fig. 5B, Supplementary Fig. 9).

## Discussion

Previous studies have favoured independent WGD events in the sturgeon and paddlefish lineages, despite their close phylogenetic relationship[8,35,38,44,45]. By accounting for the possibility of a single autopolyploidy event followed by lineage-specific rediploidization[24], our results reject independent WGDs, revealing that an ancestral WGD was followed by speciation at a time when at least 50–66% of the genome remained tetraploid. This high proportion of tetraploidy at the time of speciation provides an explanation for past studies incorrectly inferring independent WGD events[8,35,38,44,45] and has implications for acipenseriform evolution and biology.

Though not framed in the context of a shared WGD, similarity in the evolution of the sturgeon and paddlefish genomes has been previously noted[8]. For example, the six largest chromosomes (three WGD-derived chromosome pairs) appear to show particularly strong chromosomal homology between sturgeon and paddlefish (Fig. 3C)[8]; and we find that these same chromosomes have primarily undergone rediploidization before the sturgeon-paddlefish divergence. For these and other regions of the genome that have undergone rediploidization before speciation, the onset of ohnolog sequence and functional divergence (beyond allelic variation), will also have been ancestral to both extant acipenseriform lineages. This partially shared rediploidization history after shared WGD likely explains at least some of the proposed similarity in genome evolution between paddlefish and sturgeons and perhaps contributes to their remarkable ability to hybridise[8,67].

Conversely, other parts of the genome–those chromosomes and regions of chromosomes that were tetraploid when sturgeons and paddlefish diverged– rediploidized independently in each lineage, with different (and differently sized) regions, and hence different sets of genes, rediploidizing at different times. Those ohnologs that resolved from alleles after the speciation must have also undergone any sub-/neo-functionalisation/regulation independently[24]. Given that over half of the genome appears to have rediploidized after the sturgeon-paddlefish divergence, it is likely that these independently rediploidized ohnologs, and the networks they form, contribute substantially to the unique biology of each lineage. For example, a recent study of the oxytocin and vasotocin receptor (OTR/VTR) gene family, which play a variety of roles, including in social behaviour and reproduction, found that these duplicated genes emerged consistent with independent WGDs in each lineage[68]. Our results instead indicate that genes from the OTR/VTR family might be better interpreted as following the LORe model[24], having rediploidized independently in sturgeons and paddlefish.

Our estimate that at least 50–66% of the duplicated genome remained tetraploid at the point of the sturgeon-paddlefish divergence i.e. ~80 million years post-WGD, is much more drawn-out than equivalent estimates for salmonids (~60–70% rediploidization completed after ~50 million years[24,26]) and teleosts (rediploidization largely resolved after ~60 million years[13,69]) (Fig. 5, Supplementary Figs. 8 and 9). We suggest that the apparently slower evolutionary rate (in terms of both substitutions and rearrangements[7,8]) in sturgeons and paddlefish contributed to a more prolonged rediploidization period in Acipenseriformes compared to faster evolving teleosts. This may also help explain the apparent collapse of some ohnologous regions in the paddlefish genome into a single assembly sequence, although we cannot definitively exclude maintenance of tetraploidy for a very small proportion of the genome[7,26,57]. Despite this slower rate of genome rearrangement and rediploidization, and the presence of large blocks of genes sharing consistent rediploidization history in our analyses, we find ohnolog blocks that diverged before and after speciation on the same chromosomes, similar to observations in salmonids[24,26]. This appears to have often occurred through temporally isolated intra-chromosomal rearrangement events, perhaps facilitating suppression of homologous recombination, and allowing resolution of ohnologs from alleles for that genomic segment. This scenario is akin to the stepwise formation of evolutionary strata on mammalian sex chromosomes[14,58], and adds a layer of complexity atop the existing model of segmental rediploidization proposed for sturgeon[7].

Our WGD lower-bound timing (~254.7 Ma; ~241.8 Ma with fewer fossil calibrations) is substantially older than all previous estimates[7,8,38,44,45]. Firstly, previous studies have typically assumed or

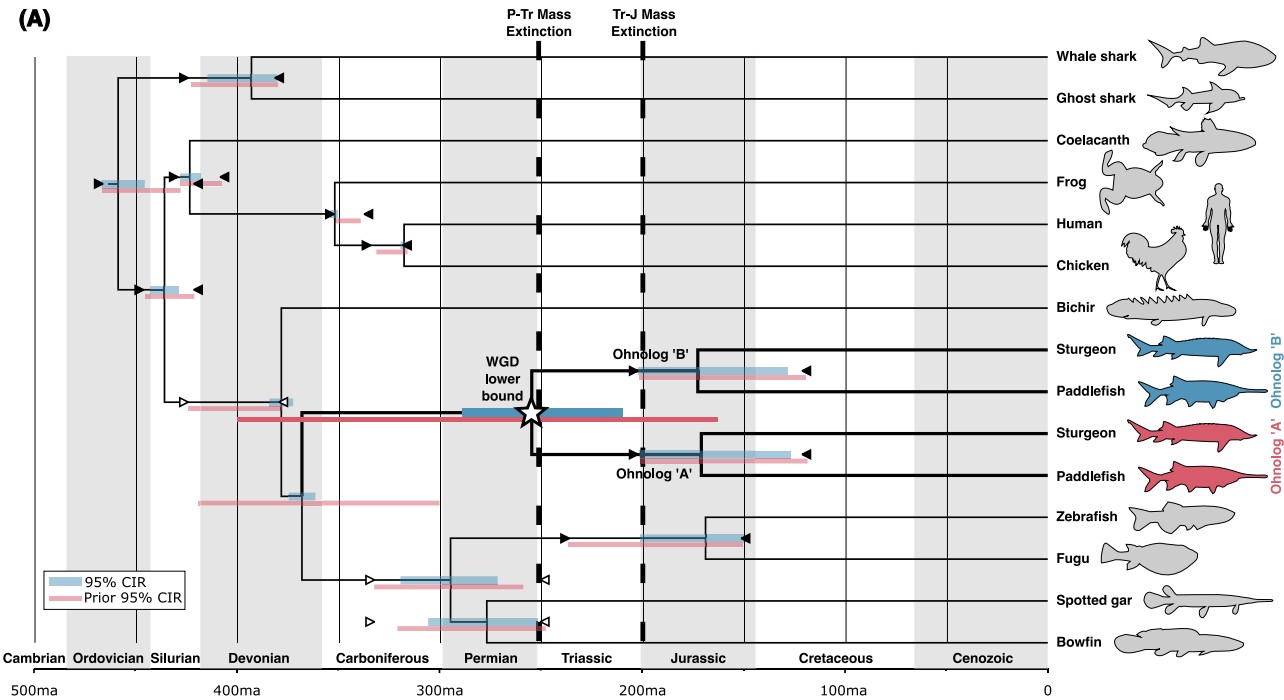

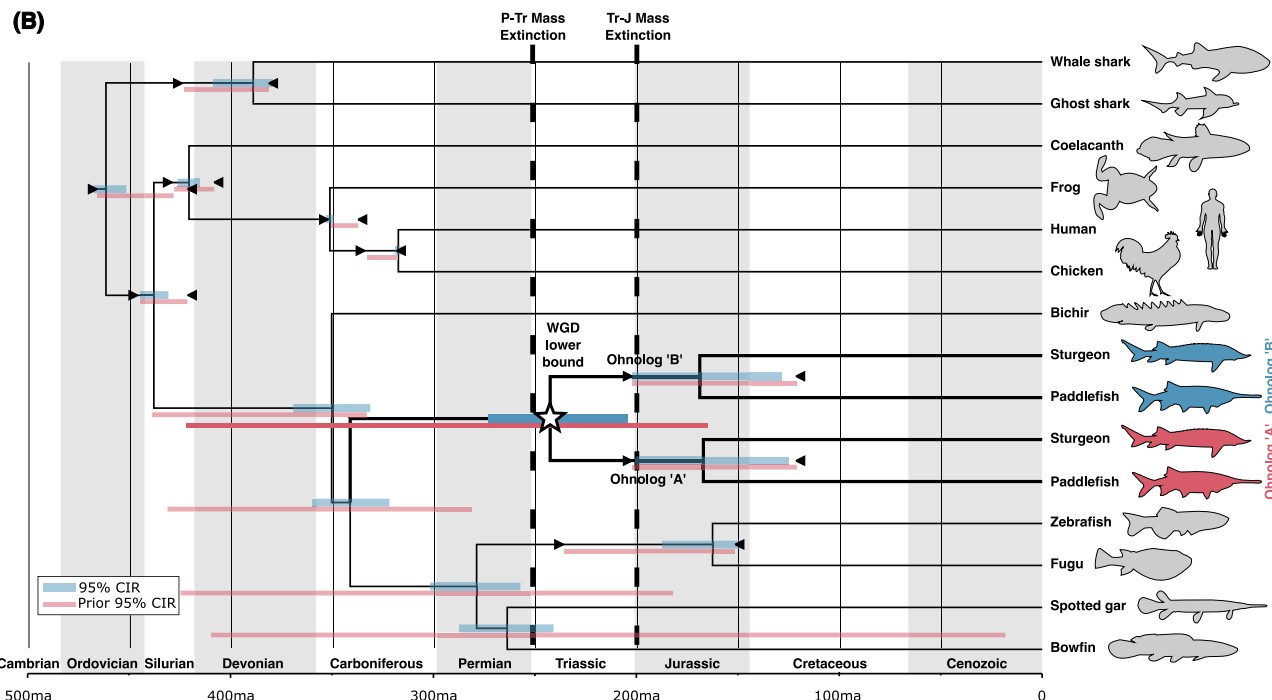

**Fig. 5 | Phylogenomic dating of the shared sturgeon-paddlefish WGD lower bound.** Ohnolog copies were randomly classified as (**A** or **B**) and concatenated for phylogenomic analysis. The jawed vertebrate Bayesian phylogenomic timetree from one of five random concatenations is shown (for all five see Supplementary Figs. 8 and 9). The 95% CIR (credibility interval) is shown for each node in blue. The 95% CIR results from an independent analysis under the prior are shown below each node in red, verifying that our priors on the WGD divergence time is sufficiently diffuse to have avoided restricting our results to the inferred WGD lower bound timing in the main analyses. Upper and lower bound fossil calibrations are shown as triangles for each calibrated divergence. Two calibration strategies were applied, the first (**A**) with more ray-finned fish calibrations (calibrations triangles with white fill are specific to this analysis) than the second (**B**), where a relaxed calibration strategy was applied to account for uncertainty in the phylogenetic placement of some ray-finned fish fossils. Individual random concatenation analyses are shown in Supplementary Figs. 8 and 9. Source data including calibrations and results are provided in Supplementary Data 1 and on figshare[106].

inferred that the WGD is not shared and that independent WGDs occurred in each lineage after the sturgeon-paddlefish divergence. Our gene tree, synteny, and $K_s$ analyses refute this and indicate that the WGD must predate the emergence of these lineages by at least long enough for ~33–50% of the genome to have rediploidized. Based on the oldest crown acipenseriform fossil, †*Protopsephurus liui* (a stem paddlefish)[70], which is dated at a minimum of ~121 Ma[71,72], this hard minimum directly contradicts all previous estimates with the exception of ~180 Ma based off the sturgeon genome analysis[7]. Furthermore, rediploidization occurring asynchronously means that common

approaches (e.g. phylogenomics or molecular clock-based analyses) used to directly estimate the absolute date of autopolyploid WGD events are unavoidably problematic, as they conflate ohnolog rediploidization time(s) with the WGD itself[24,26,62]. This means that all ohnolog pairs likely bias dating towards the present, unless they rediploidize near instantaneously after WGD, but this will be most problematic for ohnologs diverging after the sturgeon-paddlefish divergence (PostSpec). As previous studies did not consider asynchronous rediploidization, all past dating analyses will have incorporated these PostSpec ohnolog pairs in their estimation of WGD timing. We excluded these ohnolog pairs from our analyses and, unlike previous studies, estimated the WGD lower bound using a sophisticated phylogenomics approach. Thus, although our estimated WGD lower bound timing is far older than previous estimates, it should also be more accurate. Further, because of asynchronous rediploidization, it is likely still an underestimate of the true WGD timing.

Interestingly, our results support a potential role for polyploidy and asynchronous rediploidization in Acipenseriformes surviving the P-Tr and/or Tr-J mass extinction event(s). It has been proposed that WGDs in plants may confer tolerance and adaptability to extreme environmental conditions, increasing fitness in the face of mass extinction events[34,73]. Our dating of the sturgeon-paddlefish WGD lower bound suggests that the event may have occurred close to the P-Tr mass extinction period, yet our mean timing depends on calibration strategy, and confidence intervals are wide.

However, considering that most rediploidization post-dates the sturgeon-paddlefish divergence, our estimates for the WGD lower bound and of the sturgeon-paddlefish divergence are fully consistent with a model where the flexibility and functional redundancy intrinsic to a genome in the early stages of autopolyploid rediploidization contributed to the survival and success of the Acipenseriformes through at least the Tr-J mass extinction, if not also the P-Tr extinction. However, a duplicated genome is not a general shield against all extinction, and our older WGD lower bound estimates suggest it may have occurred relatively early in chondrostean evolution, implying that stem Acipenseriformes (such as †Peipiaosteidae and †Chondrosteidae)[74–77] likely split from the ancestor of extant Acipenseriformes with genomes still early in the rediploidization process.

Asynchronous rediploidization clearly exacerbates the technical difficulty of analysing WGD events—upending traditional expectations for WGD gene trees and $K_s$ plots and leading to some duplicate genomic regions being so similar as to be collapsed into a single assembly region, or perhaps even still tetraploid, long after WGD (well over 200 million years after WGD in paddlefish). In this context, through identifying distinct signals for lineage-specific rediploidization (in gene tree, synteny, and $K_s$ analyses), we provide a path forward for detecting and analysing other WGD events affected by asynchronous, lineage-specific rediploidization. We note that our framework to distinguish the gene tree distribution signal for lineage-specific rediploidization from that of phylogenetic error does not rule out alternative biological phenomena (incomplete lineage sorting or hybridisation[78]) as the source of the conflicting gene tree topology distribution. However, even the simplest scenarios required to implicate these factors rather than asynchronous rediploidization are far less parsimonious (Supplementary Fig. 10; See Supplementary Note 1 for detailed discussion).

The discovery of a mix of ancestral and lineage-specific rediploidization in both teleost[13,24,26–28] and non-teleost ray-finned fish lineages, suggests it is a general phenomenon after WGD, at least for autopolyploids. Because any individual gene cannot be considered duplicated until recombination is suppressed, such a scenario generates genomes consisting of a mosaic of shared and lineage-specific gene duplications, even though they originated from a single genome duplication. This complex relationship may help to explain the long-standing

difficulty in resolving the number and timing of WGDs in early vertebrate evolution[9,11,12,79–81].

Extensive lineage-specific rediploidization has major implications for our understanding of genome evolution following polyploidy and for our interpretation of evolution of duplicate genes including their role in adaptive evolution. This framework for the analysis and interpretation of evolution following WGD will prompt a re-examination of other autopolyploidy events, including the founding WGD at the base of all vertebrates.

## Methods
No ethical approval or other permits were required for this research.

### Ohnolog-pair datasets
In their analysis of the sterlet sturgeon genome, Du et al.[7] defined a high-confidence ohnolog pair dataset for the species, and this forms an initial basis for our analyses. To add paddlefish (GCF_017654505.1)[8] ohnologs to this dataset we used OrthoFinder[82] (v 2.5.4) to infer phylogenetic hierarchical orthogroups (PHOGs). For OrthoFinder analyses we also included a set of proteomes from species spanning jawed vertebrate phylogeny, including the ghost shark (*Callorhinchus milii*; GCF_000165045.1)[83] and whale shark (*Rhincodon typus*; GCF_001642345.1)[84] from Chondrichthyes, and human (*Homo sapiens*; GCF_000001405.39), domestic chicken (*Gallus gallus*; GCF_000002315.6), African clawed frog (*Xenopus tropicalis*; GCF_000004195.4), and coelacanth (*Latimeria chalumnae*; GCF_000225785.1) from Sarcopterygii. Within Actinopterygii we selected zebrafish (*Danio rerio*; GCF_000002035.6), fugu (*Takifugu rubripes*; GCF_901000725.2), spotted gar (*Lepisosteus oculeatus*; GCF_000242695.1)[39], and bowfin (*Amia Calva*; JAAWVP 01.1)[35] as representatives of Neopterygii, the sister group to sturgeons and paddlefishes, and grey bichir (*Polypterus senegalus*; GCF_016835505.1)[35] as their combined sister group. The longest protein sequence for each gene was used for these analyses where alternative transcripts were annotated. Default OrthoFinder settings were used, with two exceptions. First, we specified a species tree, in line with accepted jawed vertebrate relationships[7,8,35,85], to augment orthology inference: "((GhostShark,WhaleShark), ((Coelacanth,(Frog,(Human,Chicken))),(Bichir,((Paddlefish, Sturgeon),((Zebrafish,Fugu),(SpottedGar,Bowfin))))));". Second, the OrthoFinder -y flag was specified to further split PHOGs that underwent duplications after the jawed vertebrate ancestor into separate PHOGs. In post-processing of OrthoFinder results, we then performed an extra check by extracting only PHOGs including sequences from as many species as possible, while always including two sequences each from sturgeon and paddlefish. This was achieved by extracting the set of sequences descended from the most ancient ancestral species node in the OrthoFinder reconciled gene tree to not include additional sturgeon or paddlefish sequences (based on the.tsv files in the OrthoFinder Phylogenetic_Hierarchical_Orthogroups output folder). As a simple example, in an orthogroup with a gene duplication in the ancestor of Actinopterygii, with both paddlefish and sturgeon ohnologs being retained in both actinopterygian duplicates, our approach results in two resultant PHOGs, split at the level of Actinopterygii, with neither including sequences from the other duplicate or their co-orthologs from Sarcopterygii or Chondrichthyes, and both containing two sturgeon and two paddlefish sequences each.

These PHOGs were then filtered to retain only those that matched a previously inferred sturgeon high-confidence ohnolog pair[7], as well as those including at least one outgroup to allow rooting of the sturgeon-paddlefish ohnolog pair subtree, and hence inference of duplication node time relative to speciation. We also excluded gene families where both ohnologs in either paddlefish or sturgeon were present on the same chromosome, or where any paddlefish or

sturgeon sequence was present on a scaffold not assigned to one of the 60 sturgeon or paddlefish chromosomes. Lastly, we checked that the sturgeon and paddlefish sequences formed a monophyletic group in inferred PHOG gene trees (see below section on Ohnolog duplication time inference). 5439 PHOGs met these criteria (and form our ohnolog-pair set), of which 5372 also included at least one sequence each from Neopterygii and a more distantly related outgroup, all but ensuring that the two paddlefish and sturgeon sequences diverged after splitting from Neopterygii.

In all, this should have resulted in a dataset heavily enriched for ohnolog pairs in sturgeon and paddlefish. However, we note that the set includes only ohnolog pairs where both ohnologs are retained in both species and excludes PHOGs that have undergone additional duplications or losses in sturgeons, paddlefish, or their ancestral stem lineage. Similarly, it is possible that a very small subset of our ohnolog pairs may derive from complex scenarios of multiple ohnolog losses and inter-chromosomal duplications that have evaded our filters (as well as the doubly conserved synteny evidence for the original sturgeon ohnolog set) and left a pair of ohnolog look-a-likes in paddlefish and sturgeon. However, we expect PHOGs where this has occurred to be exceedingly rare in our final 5439 sturgeon-paddlefish ohnolog-pair dataset.

### Ohnolog duplication time inference

To estimate rediploidization time (i.e. duplication node time) relative to speciation for each sturgeon-paddlefish ohnolog-pair we performed phylogenetic analyses for each gene family from the pre-monophyly filtered sturgeon-paddlefish ohnolog pair PHOGs described above (5590 trees). Multiple sequence alignments were performed with MAFFT v7.487 with the --auto flag specified, and the --anysymbol flag for those datasets that included sequences containing selenocysteine (symbol U). Phylogenetic inference was performed using IQ-tree[86] (v. 2.1.4-beta COVID-edition), with the -m JTT + G flag to use the JTT[87] amino acid substitution model with four discrete gamma categories, as well as the -bb 1000 flag to specify 1000 ultrafast bootstrap replicates[48]. The resulting maximum likelihood trees were extracted for pre-processing and duplication time inference. To pre-process these trees, we used the ETE (v3) toolkit[88] python library to check that sturgeon and paddlefish sequences formed a monophyletic (in an unrooted sense) clan[89] in each PHOG gene tree, and then rooted each tree with the most distantly related sequence relative to sturgeons and paddlefish (i.e. typically ghost shark/whale shark). In a separate python script, the ETE toolkit was then used to perform strict gene tree-species tree reconciliation[88,90] to infer speciation and duplication nodes/events, before classifying (PreSpec, PostSpec, 'Other' [PreSpec-like, PostSpec-like]) and summarising the different sturgeon-paddlefish subtree gene tree topologies and frequencies recovered.

### Testing robustness of ohnolog gene tree topologies to error

To interrogate the possibility that either of the PreSpec or PostSpec gene trees derive from error, as well as to better understand the source of the 'Other' topologies we performed a suite of analyses. First, we performed unrooted AU-test[47] analyses in IQ-tree[86]. To do this we classified the three possible unrooted topologies of our four-tipped sturgeon-paddlefish subtree as either PostSpec-type or PreSpec-type based on whether they would become PostSpec(-like) or PreSpec(-like) if rooted. For each of the four rooted topology categories (i.e. Post-Spec, PreSpec, and the two 'Other' categories of PreSpec-like and PostSpec-like) we extracted the subalignment for the four sturgeon/ paddlefish sequences from each PHOG in that category, and then performed an AU-test analysis for each ohnolog-pair set specifying the three possible unrooted topologies. The frequency at which the matched unrooted topology type was not rejected for subalignments from each rooted tree topology category was then calculated and plotted.

Next, using a python script and the ETE toolkit[88] we assessed the influence that filtering our ohnolog-pair PHOG counts by increasingly higher ultrafast bootstrap percentage cut-offs (considered for the two support values within the four-tipped paddlefish-sturgeon clade only) would have on rooted topology frequencies. Starting at a cut-off of 0% and incrementing by 5%, up to 100%, we assessed the total counts and percentages of PostSpec, PreSpec, and 'Other' tree topologies recovered at each cut-off and then assessed for trends in topology frequency as the cut-off became more stringent. We also calculated and plotted the fold deviation from random (i.e. if all 15 topologies were recovered equally frequently) at which each rooted tree category was recovered for PostSpec (randomly expected 1/15 times), PreSpec (randomly expected 2/15 times) and 'Other' (randomly expected 12/15 times) topologies across the same series of ultrafast bootstrap cut-offs and evaluated the trends observed.

To assess whether gene families supporting any topology perform poorly at recovering other generally-accepted clades[49,50], and hence may be more likely to be misleading, we used a custom ETE toolkit[88] python script to assess for the monophyly of three widely-accepted clades; Tetrapoda (tetrapods; monophyly of human, chicken, and frog), Teleostei (teleost fishes; monophyly of fugu and zebrafish), and Chondrichthyes (cartilaginous fishes; monophyly of ghost/elephant shark and whale shark). This script also required that at least one sequence was present for each species in that monophyletic clade, meaning some negatives may also derive from gene loss or absence from our inferred PHOG.

To detect signs of possible systematic errors we compared a range of statistics at the sequence alignment, modelling, and inferred phylogenetic tree levels, between gene trees fitting the PostSpec, PreSpec, and 'Other' tree topologies using the two-sided Wilcox-test with Bonferroni correction in R (version 4.1.2 [2021-11-01])[91]. Alignment length and average pairwise percentage identity were calculated using the esl-alistat program from the Hmmer package [version 3.1b2; http://hmmer.org][92], while the number of parsimony informative sites[52] was extracted from IQ-tree output. PhyKIT (v. 1.11.3)[51] was used to compute the number of variable sites[52], the evolutionary rate (i.e. total tree length/number of leaf nodes)[53], treeness (i.e. sum of internal branch lengths/total tree length)[54], relative compositional variability (RCV)[54], treeness/RCV[52,54], and saturation level[55].

To further explore whether our results could derive from systematic error, we tested the use of precomputed site-heterogeneous mixture models on the set of PHOGs that had maximal support for any sturgeon-paddlefish subclade topology (ultrafast bootstrap = 100% for both support values in the sturgeon-paddlefish subtree). Specifically we tested the fit and influence the UL3[93], EX_EHO[94], and JTT + C20[95] models, which can help to alleviate systematic biases, and often fit single gene family alignments well[96,97].

### Analysis of ohnolog pairs that are single-copy in paddlefish

We analysed the set of ohnologs where both genes of the pair were present in sturgeon and only a single gene was present in paddlefish. This was performed by closely following the approach in the Ohnolog-pair datasets and Ohnolog duplication time inference sections above, but this time choosing PHOGs that contain only a single paddlefish sequence. A distinct python script, compared to that used for the main ohnolog pair analysis, was employed to classify the sturgeon-paddlefish subtree into either PreSpec-type or PostSpec-type (note that there are only three total rooted trees for this three-tip subtree, one PostSpec[-like] and two PreSpec[-like]). To better understand the distribution of PreSpec-type and PostSpec-type trees observed we considered whether the paddlefish genes in PostSpec-type trees (where the ohnologous regions are likely to be more similar, having diverged only after the sturgeon-paddlefish speciation) may in fact result from collapsing of two ohnologs into a single assembly region rather than being single-copy. To test this, we mapped paddlefish

genome DNA-sequencing reads (obtained from CNGBdb experiments CNX0162203-5, from project CNP0000867 available at https://db.cngb.org/search/project/CNP0000867/)[8] to the paddlefish genome assembly using Bowtie (v. 2.4.2). A sorted BAM file was generated for the aligned data using SAMtools (v. 1.16.1)[98]. For three separate gene sets (two-copy ohnolog pair genes [main dataset], and PreSpec-type and PostSpec-type genes that are single copy in paddlefish but form an ohnolog pair in sturgeon) gene coverage was calculated by extracting raw read depth per gene (i.e. one average value from start to end for each gene; gene coverage) from the BAM using the --by flag in mosdepth (v. 0.3.3)[99].

### $K_s$ distribution and ortholog branch length analyses

The ksd command from the wgd tool[59] was used, with default parameters, to calculate pairwise $K_s$ values for ten different ohnolog/ortholog pair datasets; eight of which were the intra-species PostSpec, PostSpec-like, PreSpec, and PreSpec-like ohnolog pairs from paddlefish and sturgeon, with the remaining two being made up of ortholog pairs from (i) the two sturgeon-paddlefish ortholog pairs from each PreSpec gene tree, and (ii) the Single-Copy Orthogroups present in all species as inferred in our OrthoFinder analysis. This latter set of Single-Copy Orthogroups could potentially incorporate some hidden ohnologs if there are cases of differential PreSpec ohnolog loss between sturgeon and paddlefish, and so is more prone to bias, but still a useful comparison. In each case the corresponding coding sequence to the amino acid sequences used in gene tree analyses were applied, and pairs where either sequence was flagged with a warning by wgd were removed and wgd reran. Pairs with $K_s$ values ≥ 0.3 were not included in the final data for visualisation. To supplement analyses of ortholog pair peak $K_s$ differences between paddlefish and sturgeon, sturgeon-paddlefish ortholog branch lengths were compared to assess for differences in amino acid evolutionary rate in substitutions per site in each species. For this analysis, we ensured that all gene trees were rooted in accordance with wider jawed vertebrate phylogeny and that sturgeon and paddlefish sequences formed a monophyletic grouping (as per Ohnolog pair datasets methods subsection above). We then extracted branch lengths for sturgeon and paddlefish from each of the two PreSpec sturgeon-paddlefish orthologs, as well as from Single-Copy Orthogroups (for which alignments and gene trees were built following the MAFFT and IQ-tree settings used in the Ohnolog duplication time inference methods subsection above) using a custom ETE toolkit python script. The sturgeon-paddlefish clade monophyly check and focus on PreSpec orthologs and (to a lesser extent) Single-Copy Orthogroups should ensure that divergence of the sturgeon and paddlefish sequence pair reflects ortholog divergence since speciation. Thus, variation in these branch lengths between species specifically captures substitution rate variation. A caveat to this exists for Single-Copy Orthogroups, where differential ohnolog loss producing hidden ohnologs could mislead analyses of this dataset. We compared the sturgeon and paddlefish branch lengths using the paired-sample Wilcox-test in R (version 4.1.2 [2021-11-01])[91]. Extreme outlier ortholog pairs with branch length values ≥ 0.15 were not included in the final data for visualisation or statistical comparison.

### Synteny analysis

The genomic coordinates of each gene in an ohnolog pair were used to anchor links between ohnologs on circos plots (drawn with circos-0.69-9[100]) of the sturgeon and paddlefish genomes. All members of an ohnolog pair were required to be present on the largest 60 chromosomes to be included. For plotting of both species in a single circos plot, PreSpec ohnolog pairs were split into separate ohnologs to be plotted as orthologs between species, while PostSpec ohnolog pairs were plotted as ohnologs within species as per the individual species plots.

### Phylogenomic divergence dating

To estimate a lower bound for the sturgeon-paddlefish WGD, we extracted the set of gene trees recovering the PreSpec sturgeon-paddlefish ohnolog pair topology with maximal support (ultrafast bootstrap = 100% for both support values in the sturgeon-paddlefish subtree). To simplify preparation for phylogenomic analysis and reduce computation time of dating analyses, we then filtered for gene families that were otherwise single copy, resulting in a set of 81 gene families. We generated five distinct datasets by randomly assigning ohnologs from a pair as the A or B copy prior to concatenation of all 81 existing gene family multiple sequence alignments. This avoids bias from a single arbitrary concatenation, while also permitting assessment of how robust results are to variations in ohnolog concatenations[62].

Upon concatenation each super-matrix was then filtered using trimAl[101] (-nogaps) and BMGE[102] (-m BLOSUM62) to trim gap-rich and saturated sites, after which 42,126 amino acid alignment sites remained in each dataset.

Phylogenomic divergence dating was then performed (on each of the five alternative super-matrices) in Phylobayes[63] (version 4.1c), specifying the site-heterogeneous CAT-GTR + G4[65] substitution model along with an autocorrelated lognormal relaxed clock model[64] (which fits and performs well in jawed vertebrate phylogenomics[85]), and a birth-death prior with soft bounds[103,104] on fossil calibrations. Fossil calibrations priors (Supplementary Data 1) were set for most nodes across the tree, with the notable exceptions of the sturgeon-paddlefish WGD lower bound timing node, and the node splitting sturgeons and paddlefish (Acipenseriformes) from Neopterygii. Calibrations, including a minimum divergence of 121 Ma[70,71] for sturgeons and paddlefish, followed Benton et al.[72], except for the lower bound on crown Chondrichthyes which was set at 381 Ma[105]. A second analysis was also performed with fewer ray-finned fish nodes calibrated in line with the difficulty of phylogenetically placing Palaeozoic fossils from this lineage[66], specifically this included the crown Actinopterygii, crown Neopterygii and crown Holostei nodes. A fixed tree topology ("(((GhostShark,WhaleShark),((Coelacanth,(Frog,(Human,Chicken))),(Bichir,(((PaddlefishA,SturgeonA),(PaddlefishB,SturgeonB)),((Zebrafish,Fugu),(Gar,Amia))))));") was specified based on accepted jawed vertebrate phylogeny[7,8,35,85] and our inference of a shared WGD, and the Chondrichthyes representatives, ghost shark and whale shark, were set as the outgroup. We verified this topology for each of our five datasets by performing a basic concatenated phylogenomic analysis in IQ-tree[86] under the JTT + G4 model[87] with 1000 UFBoot bootstrap replicates[48].

Each Phylobayes Markov chain Monte Carlo analysis was sampled for at least 10,000 cycles, with the first 5000 discarded as burn-in before calculation of inferred divergence dates and 95% credibility intervals. Runs under the prior were performed using the same settings, except for swapping to a site-homogeneous Poisson substitution model for computational efficiency since the prior over divergence times is independent of substitution model priors, to verify that the prior on the sturgeon-paddlefish WGD lower bound timing were sufficiently diffuse as to be uninformative.

### Reporting summary

Further information on research design is available in the Nature Portfolio Reporting Summary linked to this article.

## Data availability

The alignments, gene trees, random concatenation supermatrices, and phylogenomic dating chronograms generated in this study are provided on figshare (https://doi.org/10.6084/m9.figshare.19762963.v1)[106]. The phylogenomic dating node calibrations and inferred ages generated in this study are provided in Supplementary Data 1. The inferred topology categories, AU-test results, UFBoot cut-

offs, off-target clade recovery, alignment, modelling and gene tree statistics, synteny data, $K_s$ values, read depth coverage across paddlefish ohnologs, and ortholog branch length data generated in this study are provided in the Source Data file. The DNA-sequencing read data used to assess possible assembly collapse of ohnologous regions in the paddlefish genome in this study are available in the CNGBdb database under accession codes CNX0162203-5 (from project CNP0000867 available at https://db.cngb.org/search/project/CNP0000867/). Source data are provided with this paper.

## Code availability

All custom ETE3-based gene tree parsing python scripts are available on figshare (https://doi.org/10.6084/m9.figshare.19762963.v1)[106].

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

## Acknowledgements

We thank Dr. Matthias Stöck for sharing the sterlet sturgeon genome annotation. We thank Dr. Sam Giles for highlighting the need to consider the impact of excluding some fossil calibrations. A.K.R. is supported by an Irish Research Council Government of Ireland Postdoctoral Fellowship (GOIPD/2021/466). This work was supported by funding from the European Research Council, grant agreement 771419 (A.McL.).

## Author contributions

A.K.R. and A.McL. devised the study with input from D.J.M. and M.K.G. A.K.R. and D.C. carried out analyses. A.K.R. and A.McL. analysed and interpreted results. A.K.R., A.McL., D.J.M. and D.C. wrote the paper.

## Competing interests

The authors declare no competing interests.
