## [Peer Review File · Nature Communications]

Independent rediploidization masks shared whole genome duplication in the sturgeon-paddlefish ancestorReviewers' Comments:

Reviewer #1:

Remarks to the Author:

This manuscript is very interesting, with good views to analyze the sturgeon-paddlefish tetraploidy issue. In general, the overall writing is good, and numerous data were generated to support the main conclusions. However, minor revisions are required before acceptance for publication.

The authors are recommended to think about more possibilities for the ancestral WGDs. (1) The authors should consider that ILS may separate different genes into various species, which may also generate certain discrepancies between gene tree and species tree, especially after the rapid splitting of ancestral species. (2) Hybridization between different sturgeon species is a common phenomenon. During the splitting between Polyodontidae and Acipenseridae, their ancestors may have hybridized. This may cause chromosomal segmental exchanges, finally leading to biological isolation. (3) In the third case (lines 127-131), the following issue would be more reasonable: for the asynchronous WGDs, its impacts on speciation before and during these WGDs would be much stronger, although it may not cause different topologies of orthologous genes.

Lines 327-329: The authors predicted an average occurrence of the shared sturgeon-paddlefish WGD event at ~254.7 Mya, which is far away from the estimates from other research teams. It is therefore necessary to provide more explanations and discussions.

Lines 614-618: Related data should be deposited at NCBI for public availability.

Reviewer #2:

Remarks to the Author:

Redmond et al. present an interesting analysis of WGD in sturgeon and paddlefish and seek to demonstrate that the pattern of gene duplication present in the two species occurred following a WGD event prior to divergence between the two species. Previous work has suggested that the species experienced independent WGD events due to a large proportion of gene trees indicating. However, the authors here point out that this same pattern could be caused by independent and asynchronous rediploidization.

My major comment is that the crux of the paper's conclusions rest on the analysis of gene tree topologies in support of the PreSpec and PostSpec models. However, there didn't appear to be any temporal context for the analysis of genes falling into either category. For example, for all of the "other" gene tree topologies it was assumed that they followed a topology that was aligned with the two predominating models, but information about how diverged the gene copies are from one another would help to provide the needed context. I say this because, although we are dealing with old lineages, there is still a chance that deep coalescence in the ancestral tetraploid population could be affecting the gene tree topologies in some cases. I think that the way to tackle this would be to look at something like K_s for the duplicated gene pairs within species separated across the PreSpec, PreSpec-like, PostSpec, and PostSpec-like categories to see if they form roughly unimodal distributions. If they do, then I think this strengthens your argument for a shared WGD event. If there are multiple age modes for the gene pairs though, then something more complicated is going on and I think a more detailed analysis of coalescence and gene tree ages would need to be done.

Minor comments/questions:

Could loss of one of the gene copies in one of the species bias your analysis because it wouldn't be included in any of your analyses?

Figure 2 is present as part of Figure 3. I think Figure 2 can be removed since the information that it presents is aligned with the more comprehensive version in Figure 3.

REVIEWER COMMENTS

Reviewer #1 (Remarks to the Author):

This manuscript is very interesting, with good views to analyze the sturgeon-paddlefish tetraploidy issue. In general, the overall writing is good, and numerous data were generated to support the main conclusions. However, minor revisions are required before acceptance for publication.

We thank the reviewer for these kind comments.

The authors are recommended to think about more possibilities for the ancestral WGDs. (1) The authors should consider that ILS may separate different genes into various species, which may also generate certain discrepancies between gene tree and species tree, especially after the rapid splitting of ancestral species. (2) Hybridization between different sturgeon species is a common phenomenon. During the splitting between Polyodontidae and Acipenseridae, their ancestors may have hybridized. This may cause chromosomal segmental exchanges, finally leading to biological isolation. (3) In the third case (lines 127-131), the following issue would be more reasonable: for the asynchronous WGDs, its impacts on speciation before and during these WGDs would be much stronger, although it may not cause different topologies of orthologous genes.

We thank the reviewer for prompting us to consider these possible alternative explanations for the patterns we observe. We have considered these suggestions and we do not think that any of these explanations can act as an explanation for our data. We have now added a detailed 2 page supplementary text section to outline our reasons in relation to points 1 and 2 (ILS and hybridisation), aided by a new supplementary figure (Fig S10).

Unfortunately, we were unable to interpret the reviewer's meaning in point (3) in the context of our study, however we note that our third scenario does not refer to multiple asynchronous WGD events, but rather to asynchronous rediploidisation following a single ancestral WGD event. That is, different sections of the genome reverting to disomic inheritance from post-WGD tetrasomic inheritance at different times (e.g. through rearrangement events blocking homologous recombination), and in this case the majority of the genome rediploidised after speciation. We also do not address whether WGD or asynchronous rediploidisation has an impact on speciation in this work. However, we note the following that may help clear up point (3), the reviewer states that their scenario may not be expected to cause different tree topologies – thus it cannot be invoked to explain the distribution of different tree topologies we observe.

Lines 327-329: The authors predicted an average occurrence of the shared sturgeon-paddlefish WGD event at ~254.7 Mya, which is far away from the estimates from other research teams. It is therefore necessary to provide more explanations and discussions.

We have elaborated extensively on this in our discussion to better explain why our dating is appropriately much older than previous estimates. Briefly, past estimates typically relied on Ks based estimates of WGD time, whereas our dating analysis:

- 1) is based on a sophisticated phylogenomic approach that incorporates more complex evolutionary models, as well as fossil calibrations.
- 2) only relies on those genes that rediploidised prior to speciation. Past studies either considered all genes or only those rediploidising after speciation meaning that the Ks value based dates would be biased both towards the present and towards supporting WGD after speciation.

In addition to this we have added an additional phylogenomic dating analyses employing fewer fossil fish calibrations – which has previously been shown to result in younger age estimates for the major

fish lineages (Giles et al 2017)—as doing so might result in a younger estimate for the WGD lower bound. We found that even in this additional analysis the mean dating, although slightly younger, is far older than any previous estimate. This new analysis is now shown as part B of Fig. 5 alongside the original analysis that includes all calibrations (Fig. 5A), and the methods, results and discussion have been updated to reflect this.

Importantly, even though our estimate for the WGD event is much older than past studies, we still describe it as a lower bound estimate that possibly slightly underestimates the age of the WGD. This is because the ohnolog pair inferred ages (even when choosing early rediploidising ohnologs as we have here) represent their rediploidisation time rather than the time of the WGD event. In this context (and as mentioned in response to the previous comment), we have now added Ks based analyses which clearly show the limitations of not distinguishing genes based on PreSpec vs PostSpec topologies for Ks based inference of WGD time relevant to speciation.

Thus, although our dating analysis is not a perfect portrayal of the WGD time (as made clear above and in the text) it is by far the best estimate so far.

Lines 614-618: Related data should be deposited at NCBI for public availability.

The genome data we use here are previously published and cited in the methods section as appropriate. NCBI does not accept the data types that we generated such as gene trees, sequence alignments, and ohnolog pair data. However, all of these data are included as part of a FigShare repository and will have a permanent, public DOI in keeping with open data norms for the field.

We extend our thanks to the reviewer for these comments that have helped us to strengthen our manuscript.

Reviewer #2 (Remarks to the Author):

Redmond et al. present an interesting analysis of WGD in sturgeon and paddlefish and seek to demonstrate that the pattern of gene duplication present in the two species occurred following a WGD event prior to divergence between the two species. Previous work has suggested that the species experienced independent WGD events due to a large proportion of gene trees indicating. However, the authors here point out that this same pattern could be caused by independent and asynchronous rediploidization.

My major comment is that the crux of the paper's conclusions rest on the analysis of gene tree topologies in support of the PreSpec and PostSpec models. However, there didn't appear to be any temporal context for the analysis of genes falling into either category. For example, for all of the "other" gene tree topologies it was assumed that they followed a topology that was aligned with the two predominating models, but information about how diverged the gene copies are from one another would help to provide the needed context. I say this because, although we are dealing with old lineages, there is still a chance that deep coalescence in the ancestral tetraploid population could be affecting the gene tree topologies in some cases. I think that the way to tackle this would be to look at something like Ks for the duplicated gene pairs within species separated across the PreSpec, PreSpec-like, PostSpec, and PostSpec-like categories to see if they form roughly unimodal distributions.

If they do, then I think this strengthens your argument for a shared WGD event. If there are multiple age modes for the gene pairs though, then something more complicated is going on and I think a more detailed analysis of coalescence and gene tree ages would need to be done.

We initially decided not to attempt Ks-based analysis as they have attracted some criticism in the past (Tiley et al 2018) and we reasoned that diffuse Ks values expected from asynchronous rediploidisation would scramble the signal for a single WGD time 'peak'. However, we appreciate the suggestion of separately assessing the topology categories in Ks analyses. We have now included this and do find more or less unimodal distributions for each topology category. These results are consistent with what would be expected under asynchronous rediploidisation alone, where a single strong peak within a topology is not predicted. Consistent with asynchronous lineage-specific rediploidisation, we observe that Ks values are lowest for PostSpec topologies and highest for PreSpec, with speciation Ks values (i.e. sturgeon-paddlefish ortholog) falling in the middle. Intriguingly, PostSpec-like and PreSpec-like Ks values fall between their nearest 'main' topology and the speciation/ortholog peak. This indicates that at least some of the weak signal and difficulty in producing a topology for these ohnolog pairs may result from rediploidisation of these ohnologs closer in time to the speciation event. This would leave fewer substitutions with which to infer the branching order of the sequences. We have added a new results section ('Early and late rediploidising ohnologs fit distinct Ks distributions') and figure detailing these findings (Fig 4), as well as accordingly updated the methods section.

Lastly, while these Ks analyses certainly help to rule out a key role for deep coalescence driving our results, we also have now included a supplementary text and supplementary figure (Figure S10) better describing how our data cannot parsimoniously be explained by either ILS or hybridisation as opposed to asynchronous rediploidisation.

Minor comments/questions:

Could loss of one of the gene copies in one of the species bias your analysis because it wouldn't be included in any of your analyses?

Under the classic scenarios 1 and 2 presented in figure 1, a gene loss event in one of the species could not bias our analyses towards creation of a tree topology consistent with the alternative scenario (i.e. 1 or 2 of figure 1), so we are not concerned that we are observing any artefactual signal (furthermore, it would not explain the genomic arrangement of genes following a given topology we observe). Nonetheless, following the reviewer's suggestion, we tested the distribution of tree topologies for ohnolog pairs with a 2:1 sturgeon:paddlefish relationship (as our ohnolog pair dataset was based upon the ohnolog pairs initially described in the sturgeon genome and thousands of extra genes were annotated in the sturgeon genome compared to paddlefish). The results reveal a higher proportion of trees are PostSpec-type than in the main 2:2 sturgeon:paddlefish analysis (~82%; in this rooted subtree two sturgeon sequences form a clade with one paddlefish sequence as sister group).

Previous analysis of the European Grayling genome (Varadharajan et al 2018) – a salmonid that has undergone asynchronous rediploidisation—indicated that many PostSpec ('LORe' in their study) ohnologs in their assembly appear to be artefactually collapsed into a single assembly region (probably due to the challenges in distinguishing highly similar loci). To test whether this might cause the higher proportion of PostSpec-type topologies in our 2:1 sturgeon:paddlefish analysis we assessed paddlefish genome sequencing read depth coverage at each gene in the PostSpec-type and PreSpec-type sets and compared this to the values for the main 2:2 dataset genes (following Varadharajan et al 2018's approach). In line with our expectations we found that a bimodal distribution exists in the two single copy paddlefish 2:1 datasets, and this is much more pronounced for the PostSpec-type gene

set as compared to PreSpec-type. This is consistent with a small proportion of highly similar duplicate regions of the paddlefish genome being collapsed into single assembly regions. We also note that most of these 2:1 dataset genes fit the same read depth coverage distribution as 2:2 ohnologs, indicating that gene loss was also a factor in generating 2:1 sturgeon-paddlefish pairs. This is unsurprising given the well-established pattern of differential gene loss between species after WGD in general. In all we suggest that collapsing of some PostSpec-type genes may have slightly reduced the PostSpec gene tree count in our main dataset, but this would be slight and not influence the key findings of our study.

We have added a new results section ('Some paddlefish ohnologs may be collapsed into a single assembly sequence'), a supplementary figure (Fig S3) detailing the gene tree topologies, synteny and read depth for these data, and updated methods and discussion to detail this appropriately.

Figure 2 is present as part of Figure 3. I think Figure 2 can be removed since the information that it presents is aligned with the more comprehensive version in Figure 3.

We appreciate the reviewers point here. Figure 2 was initially included to show a simplistic view of the topology frequencies recovered. We have removed it and the original Figure 3 is now Figure 2.

We cordially thank the reviewer for their helpful suggestions that have led to enrichment of our manuscript.

Reviewers' Comments:

Reviewer #1:

Remarks to the Author:

The authors made careful revisions in accordance with the reviewers' comments. The present version of this manuscript is acceptable for publication.

Reviewer #2:

Remarks to the Author:

I appreciate the authors' efforts to address my previous comments and would like to thank them for their work. I think that the additions to the paper are excellent and strengthen the original arguments. I have no further comments.